# Welfare Aspects of Raising Entire Male Pigs and Immunocastrates

**DOI:** 10.3390/ani10112140

**Published:** 2020-11-17

**Authors:** Eberhard von Borell, Michel Bonneau, Mirjam Holinger, Armelle Prunier, Volker Stefanski, Susanne Zöls, Ulrike Weiler

**Affiliations:** 1Department of Animal Husbandry and Ecology, Martin-Luther-University Halle-Wittenberg, Theodor-Lieser-Str. 11, 06120 Halle, Germany; 2IFIP, The French Pork and Pig Institute, La Motte au Vicomte, B.P. 35104, 35651 Le Rheu, France; michelbonneaupro@orange.fr; 3Department of Livestock Sciences, Research Institute of Organic Agriculture FiBL, Ackerstrasse 113, 5070 Frick, Switzerland; mirjam.holinger@fibl.org; 4PEGASE, INRAE, Institut Agro, 35590 Saint Gilles, France; armelle.prunier@inrae.fr; 5Department of Behavioural Physiology of Livestock, University of Hohenheim, Schloss Hohenheim 1, 70599 Stuttgart, Germany; volker.stefanski@uni-hohenheim.de; 6Clinic for Swine, Ludwig-Maximilians-University Munich, Sonnenstrasse 16, 85764 Oberschleissheim, Germany; s.zoels@lmu.de

**Keywords:** welfare, entire males, immunocastration, behaviour, housing, management

## Abstract

**Simple Summary:**

Surgical castration of male piglets without pain treatment and anaesthesia is not only a welfare problem but also violates the integrity of the animals. The favoured alternatives of raising entire male pigs (EM) with or without immunocastration (vaccination against boar taint) may, however, impose additional welfare problems under the current housing and management conditions. This focused review is intended to summarise the current state of scientific knowledge and practical reports on critical welfare issues and risk factors. Raising EM with or without subsequent immunocastration could be a welfare conform, future-oriented alternative to conventional surgical castration of pigs, provided that they are kept in a healthy and socially stable environment with sufficient physical resources, as safeguarded by measures of enhanced animal care and management control.

**Abstract:**

For a long time, scientists assumed that newborns have a severely limited sense of pain (if any). However, this assumption is wrong and led to a “start of the exit” from piglet surgical castration. Some of the currently discussed or already implemented alternatives such as general or local anaesthesia during surgical castration raise additional welfare concerns as well as legal problems and/or are hardly applicable. The favoured long-term, welfare-friendly “gold standard” is to raise entire male pigs (EM). However, this may also impose certain welfare problems under the current conventional housing and management conditions. The specific types of behaviour displayed by EM such as mounting and aggressive behaviours but also increased exploration, which are partially linked to sexual maturation, increase the risk for injuries. The current status of knowledge (scientific literature and farmer experiences) on housing of EM suggests that environmental enrichment, space, group-stability, social constellation, feeding (diet and feeder space), health and climate control are critical factors to be considered for future housing systems. From an animal welfare point of view, an intermediate variant to be favoured to reduce problematic behaviour could be to slaughter EM before reaching puberty or to immunize boars early on to suppress testicular function. Immunization against endogenous GnRH can reduce EM-specific problems after the 2nd vaccination.

## 1. Introduction

The castration of both male and female domestic pigs has a tradition going back thousands of years. The castration of male pigs was initially carried out to influence their metabolism (increase in fat deposits) and behaviour (less aggression and sexual behaviour). Today, the main focus is on carcass quality, which means preventing the occurrence of the unpleasant boar taint and improving carcass composition with more lean meat and less fat of a desired quality. Although it is not a widespread practice, even female pigs are castrated in some regions today—as in the south of Spain—in extensive free-range husbandry to avoid unwanted pregnancies [1].

Piglets intended for fattening used to be traditionally castrated by farmers without anaesthetics during the first days/weeks of life. The surgical castration of suckling pigs can be carried out by a single person. There are differences between the methods used. The piglet is either held between the legs or placed in a restraining device. Before castration, the skin in the scrotal area can be cleaned and disinfected with a suitable antiseptic. A scalpel or side cutter is used to open the scrotum above the testicles. Either one incision is made centrally or two incisions are made parallel to the raphe scroti and the skin and tunica vaginalis are cut through. A single incision is one to two centimetres long. The testicles are placed in front of the raphe scroti with slight pressure and fixed with one hand. The spermatic cord and mesorchium are then cut through with a scalpel or an emasculator. The wound area is finally treated with an antiseptic and not closed [2].

The fact that castration of male piglets is still carried out without pain treatment and anaesthesia is based on a long-prevailing scientific view according to which newborns have a generally reduced sensation of pain and can therefore be exposed to surgical procedures without analgesia and anaesthesia. This view was based, among other things, on the discovery of the German psychiatrist and brain researcher Paul Emil Flechsig in 1872 (quoted after [3]) that nerve cells of infants are only partially myelinated and may therefore not be fully functional. It was concluded that newborn brains are not yet mature enough to feel pain. Until the 1980s it was strictly assumed that children could feel almost no pain in the first months of life. Pain reactions in babies were interpreted as a reflex. For this reason, babies were operated on with only a low dose of anaesthetic and without any pain medication [4]. It was not until 1987 that two University of Tennessee publications reported that “stress reactions” during surgery apparently decreased in infants who had been given painkillers [5,6]. Thus, one must assume that there is also a pain perception in newborns.

This idea of reduced pain perception in the first weeks of life was also applied to young animals and explains why animal welfare legislation in most countries allowed the practice of surgical castration without pain relief in the first week of life in pigs until now. The pros and cons of the various methods used for analgesia and anaesthesia during surgical castration have been reviewed extensively (see, e.g., [2,7,8,9,10,11]) and are not the focus of this publication. In brief, surgical castration with pain relief has all the advantages of surgical castration in terms of easier management, lower expression of aggressive and mounting behaviours and quality of carcasses. However, none of the various methods used nowadays guarantees that pigs are not negatively affected in terms of insufficient short- and long-term pain and stress alleviation. In addition, the loss of physical integrity of the pig and increased costs associated with labour input and pain management have to be considered.

The welfare issue around surgical castration and its alternatives has been a main driver for the promotion of raising entire males. The following condensed review summarizes the current status of knowledge on the type and frequency of welfare problems associated with housing and management of male pigs that are not surgically castrated, either with immunocastration (immunocastrates) or without (entire males, EM). Both categories of animals are very similar until the immunocastration has become effective. This is the output of the work group on Management and Housing of the COST Action IPEMA (IPEMA: Innovative Approaches for Pork Production with Entire Males). Based on the current state of scientific knowledge and practical reports, critical issues and risk factors are discussed from which recommendations on future housing and management of EM are given.

## 2. Raising of Entire Male Pigs

### 2.1. Physiological Background and Consequences

The ban of surgical castration and the fattening of EM, with or without a subsequent immunocastration, is seen as an animal friendly long-term solution which reduces stress and pain due to the surgery [12]. Another particularly attractive aspect from the point of view of resource and environmental efficiency is that uncastrated boars have a stronger protein-anabolic metabolic state due to testicular hormone production and show a reduced ad libitum feed intake [10]. Their carcasses contain more lean meat and less fat than carcasses of castrates. Feed conversion is about 10–15% better and nitrogen excretion is reduced. These effects are due to the fact that in EM, in addition to androgens, oestrogens are also produced in the Leydig cells of the testes, which both reduce protein breakdown and promote protein build-up [13]. EM are, therefore, superior in performance in terms of growth rate and feed efficiency compared to castrates and females, provided that they are kept healthy in a housing environment with all needed resources. Controlled systematic studies on lifetime health benefits of EM vs. castrates and females are not available. However, there are few studies that indicate benefits for EM pigs in health and performance pre-weaning compared to piglets that are surgically castrated [11]. The main benefit is to prevent stress and pain associated with surgical castration as reviewed in several papers. Another aspect refers to ethics, as castration and other so called “mutilations”, such as tail docking and teeth clipping, are seen as violations of the animals’ integrity and “naturalness” [14,15].

However, various problems that can occur under commercial conventional housing and management conditions were initially underestimated in their importance. Due to the increased formation of male hormones during puberty, aggressive confrontations are more frequent in EM than in castrates that may lead to more skin lesions. Particularly under husbandry conditions that are highly competitive due to a small number of feeding places or very limited access to feed, increased aggression among EM can be observed, which is more pronounced than in castrated males and females [16].

### 2.2. Welfare Aspects

Increased aggression behaviour was described in uniform groups of EM with only small differences in weight, while in heterogeneous groups—probably due to the clear weight or size specific differences in rank—fewer fights were observed [17]. Interestingly, a reduced frequency of aggression was observed when the litter siblings remained together in a stable group until slaughter (farrow-to-finish pens; [18]). In this case, there were no differences between EM and castrates. Thus, the occurrence of increased aggression in EM can be avoided by appropriate management measures.

Increased fighting activity imposes stress on all animals in a group, especially if animals are regrouped for transport and stable ranking relationships are lost [18,19,20] or if animals have to wait in the truck for a long time during or after transport to the slaughterhouse [21]. Fights and the associated stress before slaughter may also have an impact on the extent of boar taint, by increasing fat concentrations of androstenone and skatole [21].

Another aspect of EM fattening is that in the course of puberty these pigs exhibit an increased sexually motivated mounting behaviour, which lasts longer than playful or exploratory mounting (e.g., fence mounting). These mating attempts are often accompanied by intense vocalisations of the penmate being mounted, so that a reduced well-being and higher stress levels was assumed [22]. It has been observed that enriching the environment may stimulate mounting behaviour of EM even though it did not influence pubertal maturation as shown by similar plasma concentrations of oestradiol, fat concentration of androstenone and development of testes [23].

Data on the frequency of lameness and leg damage in EM vary between studies. In some studies, an increased proportion of lameness in EM groups was observed and attributed to mounting activity [19,24]. However, in other studies these problems did not occur more frequently than in females [22,25]. There are also reports that bursitis and lameness were more pronounced in castrates than in EM groups [25].

Penile injuries do not occur in castrates, as they are not able to extrude their penis. Data from post mortem analyses on incidences of either fresh wounds or scars on the penis of EM vary greatly. While Holinger et al. [26] found 3% of affected EM, Isernhagen [27] detected up to 82%, and Weiler et al. [28] reported between 64% and 95%. Mixed housing together with females and a higher age at slaughter increases the risk of penis injuries [28]. The age of the EM has a considerable influence on the number of scars in particular. The number of fresh wounds was not influenced. Serious penile injuries with suppurations and necroses were found in about 9% of the animals [28]. Again, older animals tended to be more affected, as well as young boars if they were kept together with females [28,29,30]. Comparative studies on wild boars show that penis injuries can in principle occur in the mating season even under natural conditions [28]. However, the occurrence of such injurious behaviour under natural conditions does not mean that it is also acceptable in farm animal housing as these penis injuries are presumably painful.

Previous studies have investigated the influence of group composition on sexual behaviour, but the results are not consistent. For example, Salmon and Edwards [31] describe a greater frequency of mounting in single-sex boar groups than in mixed groups. Similarly, in the studies of Rydhmer et al. [19], Boyle and Björklund [32] and Bünger et al. [33], EM in single sex groups showed significantly increased sexual activity and increased aggression behaviour compared to those in mixed-sex groups. Thus, the presence of females in the pen does not seem to stimulate the sexual behaviour of EM. A study with organically farmed pigs suggests that under enriched housing conditions, EM can be kept in single-sex groups as well as in mixed-sex groups with females without compromising their welfare [34], although these authors similarly found higher lesion scores in single-sex groups with EM, confirming earlier studies. There is evidence that welfare of female pigs is comparable between housing in mixed-sex groups with EM or with castrated males. Female pigs were rarely observed to be recipients of aggressive interactions by EM and did not have more skin lesions when compared to females in mixed groups with castrated males [34]. In addition, the risk of females being early mated by EM during fattening should be taken into account when evaluating mixed-housing systems. In the studies of Bünger et al. [33], pregnancies were found in 3% of the gilts (slaughtered at 95 kg slaughter weight). Andersson et al. [17] detected five pregnant females out of 20 (slaughtered at 107 kg live weight) when raised outdoors, but none when raised indoors to the same weight. Especially on organic farms with longer fattening periods and access to outdoor areas, such problems could potentially occur more often if kept in mixed groups. However, Holinger et al. [34] did not find any pregnancies when organic pigs were slaughtered with 92 kg slaughter weight.

In several studies, the extent of skin lesions in EM remained at a relatively low level overall, i.e., mostly only scratches and superficial abrasions were found when grading the body integument during fattening [27,33,34,35,36]. Holinger et al. [26] reported that EM performed manipulative behaviours towards pen mates more frequently, although this did not result in an apparent increase in skin, ear or tail lesions. In contrast, former studies found slightly more tail lesions in EM of around 23 kg body weight [34], more tail manipulations in females compared to castrated male pigs [37] and a decrease in tail manipulation after immunocastration of entire male pigs [38]. An explanation for the increased frequency of pen-mate manipulations in EM could be, among other factors, a feed ration that does not meet the specific demands of EM. This may be the case when fed together with females or castrates using a standard feed ration. Early warning signs for the occurrence (outbreak) of injurious behaviours (such as a sudden unusual increase in activity) as well as environmental risk factors (such as lack of substrates and feeder space, bad climate, nutrient deficiencies and diseases) may therefore be the same in EM as for pigs in general, as reviewed by D’Eath et al. [39].

Although the typical behaviour shown by EM may cause an increase in skin lesions, there is no evidence that the behaviour as well as the related noise and agitation cause a situation of chronic stress for these pigs [26,40]. Identified indicators of chronic social stress (modified fat metabolism, gastric ulcers, behavioural changes) do not differ between EM and castrated males. However, EM reacted more strongly to applied social stressors (confrontations and separations), which might imply an increased behavioural stress reactivity [26,40]. If confirmed by future studies, the latter finding could have consequences for handling or management procedures, mixing or transport of EM.

### 2.3. Consequences for Housing, Management and Transport of Entire Males

Mixing unfamiliar pigs in general, but particularly when mixing EM, increases the risk for injuries resulting from biting as a consequence of competition and formation of social hierarchy [41]. EM living in early socialized groups of different litters [41] or sibling groups [18] perform less agonistic behaviours, resulting in less skin lesion (Table 1). An early socialization could also be beneficial when split marketing is applied [42], meaning that pigs are sent to slaughter in several steps. Split marketing may induce a higher frequency of aggressive behaviour, both in EM and females [43]. However, while some publications described a comparably higher level of aggression in EM than in females [43] and slightly more skin lesions compared to castrated males and females [44], others did not find any sex specific differences in aggression [32,44] or lesions [32]. Group size and stability seem to be more important than the ratio between EM and females in lairage (holding) pens before slaughter [45], whereas mixing EM before slaughter provoked more aggression and mounting than mixing EM with females [20]. Pros and cons for different group compositions are summarized in Table 1.

The enrichment of housing as well as feeding conditions for EM have been studied to meet their presumed special demands in relation to their increased exploration activity level. However, providing EM with more space and enrichment material does not necessarily lead to a reduction in aggressive or sexual behaviour, but it reduces the risk of injuries [23,48]. This signifies that EM in an enriched pen with non-slippery flooring and enough space may be able to perform certain behaviours without risking injuries. Structuring of the pen in different areas might provide a possibility of withdrawal for lower ranking pigs. EM should therefore preferably be housed in structured pens with separate functional areas for lying, dunging and feeding and even with hiding/escape compartments which requires additional space in comparison to conventional pens with fully slatted flooring.

The frequent provision of sufficient natural material such as straw, spatially distributed on solid floor, should allow for synchronous exploration, chewing and rooting behaviour. It also reduces slipperiness of the floor and may thus contribute to the prevention of lameness. In addition, natural enrichment materials seem to have a positive effect in terms of reduced aggressive behaviour [49,50]. Enrichment may reduce the number and/or intensity of aggressions as show by a lower number of skin lesions in EM raised in an enriched environment (deep litter plus access to an outdoor concrete run) compared to EM raised on slatted flooring [23].

The provision of roughages such as grass silage or hay in addition to their regular concentrate diet may reduce feeding motivation and stress by contributing to a sustained level of satiety. It has been shown that access to grass silage reduces manipulation of pen mates both in EM and castrated males, but more pronouncedly in EM [26].

### 2.4. Gaps of Knowledge for Future Studies

Overall, it seems that the conventional fattening of young intact EM in the early stages of life prevents them from suffering stress and pain due to surgical castration, but that new animal welfare problems such as penile injuries may occur more frequently in later phases of life. It will, therefore, be necessary to examine systematically whether alternative housing and management procedures can reduce the number of penile injuries in EM housing.

Breeding approaches to provide suitable genotypes for EM production currently focus on genotypes with low androstenone and skatole levels. A decoupling of androstenone production from testicular hormone production would be particularly desirable, for example, if the enzyme CYB5B, which is involved in the synthesis of odour compounds, could be modulated [51]. However, it should be taken into account that due to the continued production of testicular hormones the behavioural problems such as increased aggressions and libido will still be present in these variants.

## 3. Sexual Maturation and Early Slaughter of Entire Male Pigs

The gender-specific aggression and sexual behaviour of EM seems to be based on the presence of sexual steroids, which are produced to a greater extent, especially during puberty (e.g., [52]). Many of the problematic behaviours of EM, such as mounting and extrusion of the penis, could be avoided by earlier slaughter, that is, before the beginning of puberty. In addition, slaughter before the onset of sexual maturity greatly reduces the formation of androstenone (e.g., [53])—an important component of boar taint. Early slaughter is the common practice to reduce boar taint in the UK and Ireland. From an animal welfare point of view, slaughter before puberty is an advantageous and easily feasible option to avoid behavioural problems associated with sexual maturation. Detailed studies are required to determine the “ideal” slaughter weight and age in this context [54]. It can be assumed that this approach also concerns sustainability factors such as environmental and resource efficiency or profitability, but the analysis of these factors is outside the scope of this review. In principle, however, it should be considered that, especially in the case of special product requirements (e.g., for ham production requiring heavy carcasses) market segmentation (sex-specific separate marketing) may be necessary, or only female slaughter pigs will be fattened to higher final weights.

Another option to decrease the behavioural problems linked to sexual maturation of male pigs would be to select lines of pigs with slightly delayed puberty so that they would be slaughtered before the occurrence of problems. Data from the literature shows that age and weight at puberty are highly heritable (heritability >0.3) in females [55,56]. No such data are available for male pigs. However, plasma oestradiol at about 110 kg liveweight, which seems to be a good indicator of pubertal development in male pigs [23], is also highly heritable [57,58].

## 4. Vaccination against Boar Taint (Immunocastration)

In immunocastration, boars are actively immunized twice against the endogenous GnRH, which is formed in the hypothalamus. For immunisation, a truncated GnRH fragment is coupled to a carrier protein to trigger an immune reaction. This coupling product, which acts as an antigen, no longer has any hormone effect itself but induces an endogenous immune reaction leading to a high level of GnRH antibodies approximately two weeks after the second vaccination [59]. The antibodies neutralise the body’s own GnRH and subsequently and interrupt the hormone cascade that controls the synthesis of steroids, such as testosterone and androstenone, in the Leydig cells of the testes [60]. According to manufacturer’s recommendations two vaccinations within an interval of at least four weeks and the second vaccination four to six weeks before slaughter should be administered. After the second immunisation, there is an initially slow increase in antibody formation within the first week, followed by a massive increase from day 5 onwards, as shown by Claus et al. [60]. As exemplified in one animal, testosterone release decreases by a factor of 10 within one day and remains at a low basal level of less than 1 ng/mL [60]. In a long-term study on immunised boars [60], it was shown that testicular function and thus the release of hormones and androstenone was safely blocked from the 8th day after the second immunization and remained inhibited for 10–24 weeks depending on the individual. Afterwards production of testicular steroids increases again and the testicular function recovers.

Several studies have investigated whether an earlier vaccination has a positive effect on the behaviour and performance of fattening pigs [30,61,62,63]. Two immunisation variants were tested in comparison to EM and surgically castrated males. The boars were immunised with either 30 and 50 kg live weight (early immunisation) or 60 and 85 kg (normal immunisation). While this early immunisation had no positive effect on the economic efficiency compared to surgical castration, problematic behaviours could be minimised [61].

A central objection to immunocastration comes from the field of workplace safety [64]. Since in humans the identical hormone GnRH is involved in the regulatory cascade of the gonads, a repeated auto-injection by the user would also lead to the formation of GnRH antibodies and to an inhibition of the pituitary gonad axis. However, this can be virtually ruled out by an adapted system of injection (see Improvac^TM^ management instructions and syringe design). In addition, the inhibition should be reversible after a few weeks as in pigs [60].

As both hormone and pheromone production are affected in this form of temporary castration, undesirable behaviours such as mounting and fighting after the second immunisation are significantly less common [10,38]. The feed intake behaviour changes dramatically due to the decrease of gonadal hormones [65,66]: feeding speed and duration per meal increase enormously, whereas the number of meals remains at the level of EM (6.7 times/day in the final fattening stage). While in this phase EM, sows and castrates eat on average 60 g of feed per minute and this value is the same for all three groups, the feeding speed of immunocastrates increases to almost 73 g/min. Differences between castrates and EM can be explained by the number of meals/day (8.6 vs. 6.5 per day), while the duration of each meal is the same (7.8 vs. 8.0 min/meal) [66]. These changes in feed intake must be taken into account when designing the diet for immunocastrates. Moreover, the high feed intake and the resulting increased fill of the digestive tract explain the poorer dressing percentage of immunocastrates compared to EM or castrates [64,67]. For a review on nutritional requirements of EM and immunocastrates see Bee et al. [68].

Schmidt et al. [35] also investigated feed intake behaviour, weight development and body damage of immunised boars in different group compositions in a performance testing station. Before the 2nd vaccination, EM spent less time visiting the feed dispenser and consumed less feed, compared to females and castrates. On the other hand, their shoulder lesion scores (as an indication for competition at the feed trough) were higher than in the other groups with females and castrates. After the 2nd vaccination, the vaccinated boars consumed significantly more feed and their daily weight gain differed significantly from those of castrates, while the damage scores no longer differed between gender. However, a separate evaluation of males in mixed-sex groups with females showed that even before the 2nd vaccination they did not show increased skin damage compared to the other groups with females and castrates. From an animal welfare point of view, it was concluded that mixed-sex housing with an early 2nd vaccination would be advantageous.

Immunization reduces but does not eliminate penile injuries [29,30]. Post mortem discovered scars may go back to the phase before the vaccination was effective. However, also fresh wounds have been discovered in penises from immunocastrated males. Another potential drawback refers to occasional non-responders to immunisation that have been reported and must be considered as a risk for unwanted early pregnancies when housed together with females [69]. Depending on the experience of the handler and the specific housing and handling constellation, vaccinations may also impose additional handling stress, especially for the 2nd or even 3rd vaccination in heavy pigs [7]. Whether the production of GnRH antibodies raises to the desired extent or whether there is an insufficient or even failed immune response depends, of course, primarily on whether the vaccinations were carried out correctly according to manufacturer’s recommendations [30,67,70]. There is a risk for incidences where the vaccine was not correctly applied (site of injection or amount) or individual animals were by mistake already marked as been vaccinated.

A recent study by Kress et al. [69] demonstrated, however, that vaccinations with Improvac™ work very reliably under various housing conditions (conventional, organic, stressful through repeated social mixing). Thus, immunocastration is effective if applied correctly to healthy individuals.

The advantages of vaccination to reduce sexual and aggressive behaviours have been reported in several studies [36,38,71,72,73,74]. However, separate housing of EM seems unavoidable due to the risk of unwanted pregnancies when housed together with females at the targeted ages or slaughter weight in most countries (at about 120 kg slaughter weight) and a relatively late suppression of testicular function in vaccinated boars (2nd vaccination). This applies particularly if sexual maturity is reached early in some individuals depending, among others, on the genotype. To our knowledge, early pregnancies were not reported in cases where females were housed together with vaccinated boars. It should, however, be considered that in some management systems with slow growing genotypes (e.g., under organic farming conditions) or heavy pig production this risk may exist. In addition, vaccinated and non-vaccinated males have different feeding requirements compared to females and castrates which would justify separate housing of males and females [36,66].

In summary, risks and benefits of boars immunised against boar taint are the same as for EM until after the 2nd vaccination when anti-GnRH antibodies are induced (usually starting within one week after the 2nd vaccination). Effectively immunised boars gradually decrease mounting and aggressive behaviours to a level similar to that of physical castrates. The extent of risks and problems to occur depend on the genotype, age and pubertal development at the time of the 2nd vaccination [35].

The immunological suppression of testicular function could, thus, represent a more animal-friendly, immediately available alternative to surgical castration, which also takes into account the consumer perspective by eliminating the formation of androstenone and skatole accumulation (see [68,75] as well as other reports in this special issue.

## 5. Conclusions

In conclusion, raising EM with or without subsequent immunocastration seems to be a welfare conform, future-oriented alternative to conventional surgical castration of pigs, provided that young intact boars are kept in a healthy and socially stable environment with sufficient physical resources, as safeguarded by measures of enhanced animal care and management control.

In particular, farmers’ experience and research have led to solutions to aggressive and mounting behaviour: (i) early socialisation in stable groups where entire males are separated from females; (ii) provision of space in structured pens; and (iii) provision of natural materials that enrich the environment of the animal.

## Figures and Tables

**Table 1 animals-10-02140-t001:** Pros and cons for different group compositions with EM (adapted from [46]).

Title	Split-Sex Groups	Mixed-Sex Groups	Early Socialized Groups	Sibling Groups
**Pro**	Targeted feeding of EM and females No early pregnancy	Common practice in GB, IRL and PT No change to existing management	Less aggressive behaviour and injuries [41] Commercially used group sizes possible	Less aggressive behaviour and injuries [18] Possibly delayed puberty and less androstenone [47]
**Con**	Additional effort for separation Separate feed formulation and storage	Earlier puberty of females [17] Insufficient utilization of EM growth potential Potential risk for early pregnancy [17,33]	Infrastructure for socialisation in the farrowing pen needed	Small groups Variation in weight at slaughter Potential risk for early pregnancy

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
