# Peer review of "Welfare Aspects of Raising Entire Male Pigs and Immunocastrates"

_animals, 2020, doi:10.3390/ani10112140_

Round 1
Reviewer 1 Report
The surgical castration of piglets is still a standard procedure in most countries to prevent boar taint that can occur in male pigs and leads to a reduction in the quality of meat. Up until now the castration procedure is carried out without (sufficient) pain relief mainly by the farmers. Already discussed or even implemented alternatives such as general or local anaesthesia raise additional welfare concerns as well as legal problems. Other alternatives such as the raising of entire male pigs or immunocastrated pigs are being explained and discussed in the present review considering their advantages and disadvantages,
Even though there are currently numerous articles on the subject, the animal welfare aspect and management requirements for the described alternatives have not yet been adequately addressed.
In the present review, the authors have dealt in detail with the environmental requirements for keeping pigs using the mentioned alternatives and the factors influencing age, sex and genetics.
All in all a well discussed and interesting contribution to scientific literature.
Table 1: Formatting error in the pdf document showing the text squeezed to the right sides of the columns.
Author Response
The authors thank the reviewer for the supportive comments.
The problem with the formatting for Table 1 has been fixed
Reviewer 2 Report
REVIEW OF MANUSCRIPT: Manuscript ID: animals-974150
Type of manuscript: Review
Title: Welfare aspect of raising entire male pigs and immunocastrates
This paper proposes to summarize the project IPEMA: Innovative Approaches for Pork Production with Entire Males. It promises to use current state of scientific knowledge and practical reports, critical issues and risk factors to discuss recommendations on future housing and management of EM. Still, I had some difficulty identifying the novelty of this paper, where it clearly goes a step beyond other published reviews on the matter. I would expect the novelty to be more clearly highlighted. This is especially lacking in the conclusions.
I had the feeling, while reading the paper, that the manuscript was finished “in a hurry”. No care to use the most appropriate literature in all instances, sometimes not citing any literature when it is clearly needed, some ideas that do not fit in the corresponding section, no good introduction and finalization of the sections (and sometimes of the paragraphs, some of which are loose statements. I am sure that with a good revision all these issues can be resolved.
General comments
- Literature cited - issues to be revised
- I found an excess of non-peer reviewed literature (most of which can be left out, in my opinion) and lack of very new peer reviewed literature (although it is perfectly acceptable to cite papers of any age, in a review one expects the new literature to be present).
- I also found an excess of passages without support from the literature. Text without citations need to be either supported or removed.
Statements that lack references are in lines 48, 50, 52, 102 (as it say “Another particularly attractive aspect…”, I conclude this is not on reference 11), 147, 202, 256.
- There is an issue to be resolved: the way authors refer to immunocastration. Immunocastrates appear to be treated in this paper as “entire until they receive the vaccine”, or a special case of entire male. I found this very intriguing. If this is what the authors think of immunocastration this needs to be clearly explained. The literature in general discusses immunocastration as a type of castration, not as a type of entire male. Just terminology, it may be argued, but considering that it diverges from mainstream, state your point of view more explicitly. However, I am not sure you convinced me that immunocastrates are entire males, regarding their behaviour and management requirements.
- Item 3.1. Welfare aspects starts and follows reviewing physiological and production aspects for some a while. This need to be resolved either changing the title or moving the contents to anappropriate section.
- Authors say “For a long time, scientists assumed that newborns have a severely limited sense of pain (if any). However, this assumption is wrong and led to a "start of the exit" from piglet surgical castration without pain relief...”
But is that the reason why castration was and continues to be done without pain relief? I would say the issue is more complex, and decisions regarding pain inflicting practices in farm animal production definitely do not depend on scientists’ beliefs.
- As the paper is presented as “the output of the work group” I would expect the corresponding publications resulting from the project cited and explicitly mentioned, as to highlight the body of new knowledge. If this is not the puropose of this review then maybe the introduction should not make such special mention to the work group?
Specific comments
Paragraph Lines 66-77 (already mentioned) - This text is very interesting, but seems out of context. It seems to blame the scientific community for the practice of castration. I don’t mean that the role of the scientific community should be excluded, but it does not explain the whole story. If you want to talk about this you should also review farmers’ and professionals’ understanding of pain and attitudes towards the issue, and the economic pressure that makes pain a side issue.
Line 114. “Another aspect refers to ethics, as castration and other so called “mutilations” such as tail docking and teeth clipping are seen as violations of the animals’ integrity and “naturalness”
This needs appropriate references to support the statement. After all, whose views are you referring to? There is specific research showing that.
Ln124 - Increased aggression behaviour
Ln 130 - Increased fighting activity
Line 138. ”so that a reduced well-being and higher stress levels must be assumed”
Who assumes that, the authors of this paper or ref 38? If the former, then move the reference to show clearly who said what.
Line 150 “greatly: While”
fix this, please
Line 153 “The age of the EM has a considerable influence on the number of scars in particular. The number of fresh wounds was not influenced. Serious penile injuries with suppurations and necroses were found in about 9 % of the animals”
I can’t identify what references this text.
Ln 160 Do you imply that sexual behaviour is a welfare problem? This makes me wonder if the section should be “behaviour and welfare problems” (still consider that the beginning of the section is neither and needs to be relocated).
Ln 180 this sentence is a paragraph? And also it seems to be totally out of context because the next paragraph returns to the same issue of lesions. That, also needs to be relocated to where you were discussing this issue, instead of bringing the same issue again.
Line 187 “An explanation for the increased frequency of pen-mate manipulations in EM could be a feed ration that does not meet the specific demands of EM.”
Do you really think that? D’Eath’s reference does not support this. The reasons for penmate directed behaviours are quite complex, and certainly related to the environment, less clearly to “nutrition” factors.
Line 194 “Although the typical behaviour shown by EM may cause an increase in skin lesions, there is no evidence that the behaviour as well as the related noise and agitation cause a situation of chronic stress for these pigs”
If there is no evidence why do you say this?
Ln 207 - why do you compare EM with females and not with castrate males?
Table 1 is very difficult to assess given the format. I don’t think it is relevant enough to be in the paper unless more is argued in the text about group formation.
Ln 212 “Providing EM with more space and enrichment material does not necessarily lead to a reduction in aggressive or sexual behaviour, but it reduces the risk of injuries [22,46].”
Do these references with EM pigs?
Line 220 “The frequent provision of sufficient natural material such as straw, spatially distributed on solid floor, should allow for synchronous exploration, chewing and rooting behaviour”
This needs to be supported by reference.
Also, you say that it “also reduces slipperiness”. What else were you referring to?
Line 228 and 232 Two paragraphs that are statements, not exactly a paragraph.
The last 2 paragraphs of the section do not refer to “3.2. Consequences for housing, management and transport of entire males”. The first is a concluding remark and the last one is lost there. Please organize the section.
Ln 249 I don’t see what idea in the sentence the reference 50 supports.
Ln 254 “From an animal welfare point of view, slaughtering before puberty is an easily feasible way” what do you mean feasible from an animal welfare point of view?
Line 262 But what you say here does not fit in “early slaughter”.
Maybe the title of the section could be adapted, something signalling at puberty and slaughter rather than early slaughter.
Line 274 The paper cited is not the most appropriate reference here, you need one that shows the mode of action of the vaccine, not its efficacy. Either the review cited earlier or an earlier paper describing the technique.
Line 276 Same comment, not the best reference as this one focuses on the behavioural effects and does not even measure androstenedione.
Line 282 reference format
Line 283 In a long-term study on immunised boars [59,60]
One study and 2 references? Do we need the reference in German? It does not help much...
Line 289 What do these references say? They raise the issue of the vaccination not being done correctly according to manufacturer’s recommendations? Please be more specific in this sentence.
Line 291 “is the subject of ongoing investigations [63].”
Can you elaborate? Or delete comment that, as such, does not add information.
Line 293 "Two immunisation variants were tested in comparison to EM and surgically castrated males. The boars were immunised with either 30 and 50 kg live weight (early immunisation) or 60 and 85 kg (normal immunisation). While this early immunisation had no positive effect on the economic efficiency compared to surgical castration, problematic behaviours could be minimised. "
which reference is that? Where can it be found?
Ln 302 - “In addition, the inhibition is reversible after a few weeks as in pigs”
Do you have a reference for this? If not, change the sentence to a suggestion instead of statement.
Ln 304 The paragraph starts about one issue (behaviour) and finishes with something not related (feeding). Please connect the ideas more explicitly.
Ln 313 - "Moreover, the high feed intake and the resulting increased fill of the digestive tract explain the poorer dressing percentage of immunocastrates compared to EM or castrates" -
Are there welfare implications of high feed intake and the resulting increased fill of the digestive tract?
Line 316 This paragraph seems to be “and many other issues also to be mentioned”.
They are not connected. Please make this a paragraph that makes more sense, something like “besides, other criticisms, or problems, or issues to be resolved… and then you can list these unrelated things.
These ideas need to be discussed together:
Ln 323: "Another objection from pork industry is that a high rate of non-responders may exist. A recent study by Kress et al. [70] demonstrated, however, that vaccinations with Improvac™ work very reliably under various housing conditions (conventional, organic, stressful through repeated social mixing). Thus, immunocastration is effective if applied correctly to healthy individuals." -
Line 287: “Whether the production of GnRH antibodies raises to the desired extent or whether there is an insufficient immune response depends, of course, primarily on whether the vaccinations were carried out correctly according to manufacturer’s recommendations [29,61,62]. The extent to which stress and other environmental factors lead to vaccination failure even if the vaccination is carried out correctly 290 is the subject of ongoing investigations [63].”
Ln 323 - "Another objection from pork industry is that a high rate of non-responders may exist. A recent study by Kress et al. [70] demonstrated, however, that vaccinations with Improvac™ work very reliably under various housing conditions (conventional, organic, stressful through repeated social mixing). Thus, immunocastration is effective if applied correctly to healthy individuals." -
I suggest to move to the paragraph of ln 287.
Ln 327 This paragraph is also somewhat confusing. Feed intake had already been covered, and then you bring in the same paragraph another issue.
Ln 339 “However, separate housing of EM seems unavoidable due to the risk of unwanted pregnancies”
But are you talking about immunocastration or EM? Are unwanted pregnancies a problem in immunocastration to the point of requiring separate housing? Please support this with references.
Line 344 “In addition, vaccinated and non-vaccinated males also have different feeding requirements compared to females and castrates”
Again, not clearly or obviously related to the rest of the paragraph.
Line 346 But what is it, EM and immunocastration are the same or effectively immunized pigs are like castrated pigs?
Line 354 “See other reports in this special”, I suggest you cite the specific papers you think that support this statement.
Conclusion:
This conclusion falls short to the objectives proposed in the introduction (and the IPEMA project I would say) similar things were said in previously published reviews… I would expect some exiting conclusions and recommendations like those presented in the seminar of conclusion of the project.
Author Response
Title: Welfare aspects of raising entire male pigs and immunocastrates
The authors thank the reviewer for detailed and very helpful suggestions to improve the manuscript. Please note, that the line numbers now refer to the revised manuscript
This paper proposes to summarize the project IPEMA: Innovative Approaches for Pork Production with Entire Males. It promises to use current state of scientific knowledge and practical reports, critical issues and risk factors to discuss recommendations on future housing and management of EM. Still, I had some difficulty identifying the novelty of this paper, where it clearly goes a step beyond other published reviews on the matter. I would expect the novelty to be more clearly highlighted. This is especially lacking in the conclusions.
Indeed, the reviewers’ impression of a lack in novelty of this paper is understandable. However, the authors did a comprehensive search on current state of knowledge regarding publications and practical reports, which are to our knowledge all included. A very important goal of IPEMA was to disseminate the knowledge to stakeholders and a webinar was organised on 15th September 2020 to this effect. The present paper is meant to offer more detailed information to those stakeholders who want to go deeper in the available science-based information. Therefore, the main aim is to provide a comprehensive view rather than to focus on the new knowledge.
I had the feeling, while reading the paper, that the manuscript was finished “in a hurry”. No care to use the most appropriate literature in all instances, sometimes not citing any literature when it is clearly needed, some ideas that do not fit in the corresponding section, no good introduction and finalization of the sections (and sometimes of the paragraphs, some of which are loose statements. I am sure that with a good revision all these issues can be resolved.
Our paper was not finished in a hurry as every author had sufficient time to work on the manuscript. The reviewer is right in that not every single statement is backed up by a reference. This has in part to do with some practical experiences that are not visible in the scientific literature or aspects for improvement of housing and management that were transferred from other problem areas such as tail biting in which the preventive aspects (such as enrichment and social stability) might be similar for prevention of unwanted behaviours in entire males. This has been however clearly stated in the paper. We are very willing to improve the text by answering questions and following recommendations, but it is sometimes difficult for us because they are not specified in the report.
General comments
- Literature cited - issues to be revised
- I found an excess of non-peer reviewed literature (most of which can be left out, in my opinion) and lack of very new peer reviewed literature (although it is perfectly acceptable to cite papers of any age, in a review one expects the new literature to be present).
We did our best to include peer-reviewed outcomes from members of our IPEMA group and other researchers, which are also included in the current special issue of animals. A focused review should also refer to previous reviews or reports in which the welfare aspects of castration in general and its alternatives were handled. These previous reviews were not specifically targeted towards entire males as done in the present review. We have included about 25 references that were published between 2015 and 2020.
- I also found an excess of passages without support from the literature. Text without citations need to be either supported or removed.
We have checked our statements regarding references and included them if necessary or removed them if there is no indication at all to support them (see details below)
Statements that lack references are in lines 48, 50, 52, 102 (as it say “Another particularly attractive aspect…”, I conclude this is not on reference 11), 147, 202, 256.
A reference for statements in l 48, 50, 52 has been added (EFSA Report 2004, [1]). The aspect of resource and environmental efficiency refers to reference [10]. We have included this reference after our statement in l 106.
The statement in l 147 has been taken out
L 202: reference [41] was added at the end of the sentence (line 204)
L 256: this “ideal” slaughter weight or age refers to a lack of knowledge in the context of consequence for early slaughter > “Detailed studies are required….” We have, however, indicated that this is out of scope for this review. A reference to that issue is given in the EU-Report on best practices for production, processing and marketing of meat from EM (2019) that we have added as a new reference [54], line 262.
- There is an issue to be resolved: the way authors refer to immunocastration. Immunocastrates appear to be treated in this paper as “entire until they receive the vaccine”, or a special case of entire male. I found this very intriguing. If this is what the authors think of immunocastration this needs to be clearly explained. The literature in general discusses immunocastration as a type of castration, not as a type of entire male. Just terminology, it may be argued, but considering that it diverges from mainstream, state your point of view more explicitly. However, I am not sure you convinced me that immunocastrates are entire males, regarding their behaviour and management requirements.
The authors are convinced that immunocastrates can be considered as entire males until they are treated with the second anti-GnRH vaccination. Therefore, it is feasible to treat them as EM until immunocastration has become effective. There are numerous publications that indicate that immunocastrates show a similar behaviour as EM up to that time. It is therefore justifiable to treat them as EM until the second vaccination, after which they have to be evaluated differently to EM. Therefore, “risks and benefits of boars immunised against boar taint are the same as for EM until after the 2nd vaccination when anti-GnRH antibodies are highly produced (usually starting within one week after the 2nd vaccination).” Actually, although we call it immunocastration (not everyone of us favours this term), it should not be scientifically considered as another form of castration, as it is reversible and just suppresses the testicle function for a defined period of time. However, since there was some ambiguity, the introduction has been modified (lines 92-94) so that what we mean by entire males and immunocastrates becomes clearer: “frequency of welfare problems associated with housing and management of male pigs that are not surgically castrated, either with immunocastration (immunocastrates) or without (entire males, EM). Both categories of animals are very similar until the immunocastration has become effective”.
- Item 3.1. Welfare aspects starts and follows reviewing physiological and production aspects for some a while. This need to be resolved either changing the title or moving the contents to anappropriate section.
Welfare encompasses a much broader range of aspects than just behaviour and health, namely indications of stress and inappropriate biological functioning (see e.g. Fraser, 2003: “Assessing Animal Welfare at the Farm and Group Level: The Interplay of Science and Values). The first paragraph of chapter 2.1 is intended to describe the physiological background in the development of EM responsible for potential benefits or drawbacks of not surgically castrated as a piglet, even inclusion of ethical considerations such as naturalness or completeness. We have therefore included a new sub-headline for this background information.
- Authors say “For a long time, scientists assumed that newborns have a severely limited sense of pain (if any). However, this assumption is wrong and led to a "start of the exit" from piglet surgical castration without pain relief...”
But is that the reason why castration was and continues to be done without pain relief? I would say the issue is more complex, and decisions regarding pain inflicting practices in farm animal production definitely do not depend on scientists’ beliefs.
The reviewer is right that the issue is more complex than just referring to the aspect of limited sense of pain. However, as stated in the introduction, there are of course other aspects besides less aggression and sexual behaviour, such as “carcass quality, which means preventing the occurrence of the unpleasant boar taint and improving carcass composition with more lean meat and less fat”. On the other hand, it is obvious that the main move towards alternatives to surgical castration such as pain treatment or raising of EM with or without immunocastration comes from the scientific evidence that even newborn piglets feel pain and that castration without pain relief is no longer acceptable.
- As the paper is presented as “the output of the work group” I would expect the corresponding publications resulting from the project cited and explicitly mentioned, as to highlight the body of new knowledge. If this is not the puropose of this review then maybe the introduction should not make such special mention to the work group?
The objective of this special issue, in which this paper is intended to belong to, was to collect all outputs of the working groups but also from corresponding papers that fit into the issue. We are not aware of any other corresponding publication not mentioned in this paper or others in this special issue of animals. A summary of all IPEMA outputs is also accessible via the website of this Cost-Action. It should be noticed that COST-Actions do not support any original research. The disseminations are based on interpretations of the current state of knowledge.
Specific comments
Paragraph Lines 66-77 (already mentioned) - This text is very interesting, but seems out of context. It seems to blame the scientific community for the practice of castration. I don’t mean that the role of the scientific community should be excluded, but it does not explain the whole story. If you want to talk about this you should also review farmers’ and professionals’ understanding of pain and attitudes towards the issue, and the economic pressure that makes pain a side issue.
The authors do not intend to blame the scientific community for the practice of castration. We are aware that this has been done for practical and economic reasons for a long time. As this review deals with welfare aspects, it is expected that we evaluate these aspects from a scientific point of view. It is not our intention to elaborate on the consequences of various scenarios for farmers, at least economically. Nevertheless, we suggest changes/adaptations in the management as a consequence of raising EM and immunocastrates.
Line 114. “Another aspect refers to ethics, as castration and other so called “mutilations” such as tail docking and teeth clipping are seen as violations of the animals’ integrity and “naturalness”
This needs appropriate references to support the statement. After all, whose views are you referring to? There is specific research showing that.
Ethical considerations are of course not based on hard core scientific data. These views are of philosophical nature and also play an important role in consumer perception/acceptance of farm animal procedures. We have given references for these ethical aspects [14, 15] but we could add more regarding the concept of animal integrity and naturalness.
Ln124 - Increased aggression behaviour
We have checked the references and found them appropriately cited for this statement
Ln 130 - Increased fighting activity
We have checked the references and found them appropriately cited for this statement
Line 138. ”so that a reduced well-being and higher stress levels must be assumed”
Who assumes that, the authors of this paper or ref 38? If the former, then move the reference to show clearly who said what.
We have cited the interpretation of screaming by mounted pigs from this paper [22]: “loud and high pitched screaming in pigs is an indicator of stress and reduced welfare” We have changed it into “was assumed” instead of “must be assumed”
Line 150 “greatly: While”
fix this, please
Done
Line 153 “The age of the EM has a considerable influence on the number of scars in particular. The number of fresh wounds was not influenced. Serious penile injuries with suppurations and necroses were found in about 9 % of the animals”
I can’t identify what references this text.
This number referred to paper [28] and was added as a reference at the end of the sentence (line 156).
Ln 160 Do you imply that sexual behaviour is a welfare problem? This makes me wonder if the section should be “behaviour and welfare problems” (still consider that the beginning of the section is neither and needs to be relocated).
We do not consider sexual behaviour per se as a welfare problem. We think that only the consequences under current housing and management condition may impose welfare problems such as excessive mounting and injurious behaviour. Therefore, we cannot exclude sexually motivated behaviour out of this problem context
Ln 180 this sentence is a paragraph? And also it seems to be totally out of context because the next paragraph returns to the same issue of lesions. That, also needs to be relocated to where you were discussing this issue, instead of bringing the same issue again.
This was not meant to be a paragraph. We have moved the sentence to l 214 before Table 1.
Line 187 “An explanation for the increased frequency of pen-mate manipulations in EM could be a feed ration that does not meet the specific demands of EM.”
Do you really think that? D’Eath’s reference does not support this. The reasons for penmate directed behaviours are quite complex, and certainly related to the environment, less clearly to “nutrition” factors.
This explanation is actually coming more from practical farm experience with EM saying that EM may need adapted diets (as enrichment to manipulate) that keep them calmer, but we do not have a reference for this. As you said, we propose to modify the sentence saying, that nutrition among other factors could contribute to the problem. Indeed, this is not supported by the reference of D’Eath.
Line 194 “Although the typical behaviour shown by EM may cause an increase in skin lesions, there is no evidence that the behaviour as well as the related noise and agitation cause a situation of chronic stress for these pigs”
If there is no evidence why do you say this?
We have stated that, as measures of cortisol and other indicators of chronic stress do not support that they are in a situation of chronic stress (see reference 26 and 40). We added the two references directly after this statement (line 197).
Ln 207 - why do you compare EM with females and not with castrate males?
The reason is that the paper [43] we are referring to only included EM and females. Paper 44 compared EM with castrated males and females. This was added to the text before the reference: “compared to castrated males or females”
Table 1 is very difficult to assess given the format. I don’t think it is relevant enough to be in the paper unless more is argued in the text about group formation.
Sorry, there was a problem of the lay-out. Now it is solved and the table should be easier to assess. We think that it is important to give a reference to a practical guide on the pros and cons of different group compositions as developed for organic farming.
Ln 212 “Providing EM with more space and enrichment material does not necessarily lead to a reduction in aggressive or sexual behaviour, but it reduces the risk of injuries [22,46].”
Do these references with EM pigs?
Study [23] was done with EM and castrates were compared to EM in [47]
Line 220 “The frequent provision of sufficient natural material such as straw, spatially distributed on solid floor, should allow for synchronous exploration, chewing and rooting behaviour”
This needs to be supported by reference.
This is something very well known that does not need a reference to support, if pigs have more material to explore, they will use it for exploration, chewing and rooting together. It is obvious to anyone (farmers, researchers, students…) who has worked with pigs. This is “common knowledge”. If we were writing a review on thermoregulation, we may have written “When animals are too hot, they move to a cooler place, if available” without giving any reference to support it.
Also, you say that it “also reduces slipperiness”. What else were you referring to
I think we can presume that the frequent provision of straw can reduce slipperiness without coming up with another reference for this Again, it is “common knowledge” that does not need a reference to support. With “also” we mean additionally to the above mentioned beneficial effects.
Line 228 and 232 Two paragraphs that are statements, not exactly a paragraph.
The second paragraph, although just one sentence, refers to another effect of housing constellation on the behaviour at the abattoir. Therefore, it had to be separated from the previous one dealing with feeding behaviour
The last 2 paragraphs of the section do not refer to “3.2. Consequences for housing, management and transport of entire males”. The first is a concluding remark and the last one is lost there. Please organize the section.
The last two paragraphs of this section are concluding remarks on gaps of knowledge regarding control of penis biting and genetic approaches to tackle problem behaviours. We have introduced this paragraph by a new sub-heading “Gaps of knowledge for future studies”.
Ln 249 I don’t see what idea in the sentence the reference 50 supports.
Yes, indeed, it would be sufficient to refer only to reference [52]
Ln 254 “From an animal welfare point of view, slaughtering before puberty is an easily feasible way” what do you mean feasible from an animal welfare point of view?
“advantageous and easily feasible option” might be better suited instead. See changes made
Line 262 But what you say here does not fit in “early slaughter”.
Maybe the title of the section could be adapted, something signalling at puberty and slaughter rather than early slaughter.
Ok, we modified the title of this chapter in “Sexual maturation and early slaughter”
Line 274 The paper cited is not the most appropriate reference here, you need one that shows the mode of action of the vaccine, not its efficacy. Either the review cited earlier or an earlier paper describing the technique.
Line 276 Same comment, not the best reference as this one focuses on the behavioural effects and does not even measure androstenedione.
Ok, this was not correctly assigned to the appropriate reference. It actually refers to [60], this has been adapted.
Line 282 reference format
Ok, it should be [60] > changed
Line 283 In a long-term study on immunised boars [59,60]
One study and 2 references? Do we need the reference in German? It does not help much...
You are right, we do not need a 2nd German reference for that. It was deleted.
Line 289 What do these references say? They raise the issue of the vaccination not being done correctly according to manufacturer’s recommendations? Please be more specific in this sentence.
Yes, that is what we meant, as failures to apply the vaccine correctly or missing animals (e.g. they were by mistake already marked as been vaccinated) can lead to non-responders. This argument has been raised by opponents of the procedure. We have included more details on that and moved this issue to the appropriate paragraph following l 341
Line 291 “is the subject of ongoing investigations [63].”
Can you elaborate? Or delete comment that, as such, does not add information.
You are right, we do not need to refer to that conference contribution any more as this was published in the meantime, see [69]. The sentence should be deleted as we actually refer to the new paper later on: “A recent study by Kress et al. [69] demonstrated, however, that vaccinations with Improvac™ work very reliably under various housing conditions (conventional, organic, stressful through repeated social mixing).”
Line 293 "Two immunisation variants were tested in comparison to EM and surgically castrated males. The boars were immunised with either 30 and 50 kg live weight (early immunisation) or 60 and 85 kg (normal immunisation). While this early immunisation had no positive effect on the economic efficiency compared to surgical castration, problematic behaviours could be minimised. "
which reference is that? Where can it be found?
The latter statement refers to the study of Andersson et al. 2012 [61]. We have added this reference after this sentence
Ln 302 - “In addition, the inhibition is reversible after a few weeks as in pigs”
Do you have a reference for this? If not, change the sentence to a suggestion instead of statement.
This is clearly reversible as reported in the reference of Claus, R., Rottner, S., Rueckert, C., 2008. Individual return to Leydig cell function after GnRH-immunization of boars. Vaccine 26, 4571-4578. This reference was added [60].
Ln 304 The paragraph starts about one issue (behaviour) and finishes with something not related (feeding). Please connect the ideas more explicitly.
Both aspects refer to changes in behaviour, sexual and feeding behaviour. We just point out, that this might influence diet formulation and meat quality which is out of scope for this paper (we referred to the paper of Bee et al. (Animals 2020, 10(11), 1950, [68]) in this special issue.
Ln 313 - "Moreover, the high feed intake and the resulting increased fill of the digestive tract explain the poorer dressing percentage of immunocastrates compared to EM or castrates" -
Are there welfare implications of high feed intake and the resulting increased fill of the digestive tract?
Indeed, this could even have a positive effect regarding their welfare, but this has not been explicitly studied
Line 316 This paragraph seems to be “and many other issues also to be mentioned”.
They are not connected. Please make this a paragraph that makes more sense, something like “besides, other criticisms, or problems, or issues to be resolved… and then you can list these unrelated things.
Indeed, this paragraph has been modified accordingly and issues interconnected. The last sentence has been moved to the paragraph before Table 1 that deals with social hierarchy and stability following l 211. An introduction to the following paragraph after the table that deals with enrichment has been added starting in l 219
These ideas need to be discussed together:
Ln 323: "Another objection from pork industry is that a high rate of non-responders may exist. A recent study by Kress et al. [70] demonstrated, however, that vaccinations with Improvac™ work very reliably under various housing conditions (conventional, organic, stressful through repeated social mixing). Thus, immunocastration is effective if applied correctly to healthy individuals." -
Line 287: “Whether the production of GnRH antibodies raises to the desired extent or whether there is an insufficient immune response depends, of course, primarily on whether the vaccinations were carried out correctly according to manufacturer’s recommendations [29,61,62]. The extent to which stress and other environmental factors lead to vaccination failure even if the vaccination is carried out correctly 290 is the subject of ongoing investigations [63].”
These two paragraphs have been merged together (line 341)
Ln 323 - "Another objection from pork industry is that a high rate of non-responders may exist. A recent study by Kress et al. [70] demonstrated, however, that vaccinations with Improvac™ work very reliably under various housing conditions (conventional, organic, stressful through repeated social mixing). Thus, immunocastration is effective if applied correctly to healthy individuals." -
I suggest to move to the paragraph of ln 287.
We did it the other way around and moved the paragraph following l 287 in front of the statement starting in l 323 (new line 341). We do not necessarily need to mention that this is an objection from the pork industry as this might also be criticized by others. We have indicated before that this could be a potential drawback
Ln 327 This paragraph is also somewhat confusing. Feed intake had already been covered, and then you bring in the same paragraph another issue.
Ok, you are right. We have moved this paragraph to the appropriate place following l 323 and rephrased the sentence in l 363
Ln 339 “However, separate housing of EM seems unavoidable due to the risk of unwanted pregnancies”
But are you talking about immunocastration or EM? Are unwanted pregnancies a problem in immunocastration to the point of requiring separate housing? Please support this with references.
This refers primarily to EM but also to immunocastrates if vaccinated late. We do not have references for early pregnancies in mixed sex housing of females and immunocastrates. We have stated here, that there is a risk for that if boars are vaccinated at a very late age/weight, as it might happen under organic farming conditions or in heavy pig production systems. We have modified this paragraph accordingly.
Line 344 “In addition, vaccinated and non-vaccinated males also have different feeding requirements compared to females and castrates”
Again, not clearly or obviously related to the rest of the paragraph.
This sentence has been modified to highlight other aspects (feeding) to justify separation of sexes (line 373)
Line 346 But what is it, EM and immunocastration are the same or effectively immunized pigs are like castrated pigs?
Obviously, EM and immunocastrates before their 2nd vaccination can be viewed similar in terms of their behaviour as well as other aspects. Even after the 2nd vaccination they do not necessarily behave like castrates. This has to do with learned behaviours during their development and only gradual changes due to the ongoing suppression of their sexual function. It is also known that their feeding behaviour is very different from castrates, but this would be more an issue handled by the other review dealing with feeding and nutritional requirements.
Line 354 “See other reports in this special”, I suggest you cite the specific papers you think that support this statement.
Yes, we should do that. But we first need to have all these papers in line to give the appropriate reference as some of them are still under revision (> like the one of the WGs on Genetics). At this time point we could refer to the papers of Aluwé et al. and Bee et al. [68, 75] and pinpoint to all other reports of IPEMA members in this special issue
Conclusion:
This conclusion falls short to the objectives proposed in the introduction (and the IPEMA project I would say) similar things were said in previously published reviews… I would expect some exiting conclusions and recommendations like those presented in the seminar of conclusion of the project.
Indeed, the conclusion is rather short and not much detailed. What do you mean by exciting? A review summarises the state of knowledge. We do not report any original research with new innovative approaches. The issue is also too complex for suggesting detailed “golden standards” for future housing and management. As suggested we have in addition included more detailed recommendations as were given under “Practical solutions for EM production” in the press release of the IPEMA seminar conclusions:
In particular, farmers' experience and research have led to solutions to aggressive and mounting behaviour: i) early socialisation in stable groups where entire males are separated from females; ii) provision of space in structured pens; iii) provision of natural materials that enrich the environment of the animal.
Reviewer 3 Report
This is a clear overview of considerations for moving forward to stopping surgical castration in male piglets. I have only some minor comments.

Author Response
The authors thank the reviewer for the supportive comments
l L 22; we added "(vaccination against boar taint)" to explain what immunocastration means. As it is within the simple summary, we can't explain the procedure in all details referring to the temporal suppression of testicular function. This is done later in the manuscript.
L 31: we deleted "without pain relief" as the ban declaration was indeed related to castration in general
L 206: changed from "slaughtering" to "slaughter" throughout the manuscript
The formatting problem in the pdf version of Table 1. has been fixed
L 254: The question of boar taint incidences in the UK and Ireland is difficult to answer as it can be expected to be much lower due to earlier stage of maturity. To our knowledge, representative surveys on this prevalence in UK and IRL are not available. A report from an Irish olfactory panel (Allen et al. 2001) found 8 % of the tested pigs with a boar taint problem. Another problem is that we do not have a uniform objective and reproducible method for boar taint assessment among different regions and countries. This issue is out of our scope and discussed in other papers within this special issue.
Round 2
Reviewer 2 Report
I thank the authors for reflecting on my comments and incorporating several of my suggestions. I understand that some of my comments were interpreted by the authors as my opinion, and of course, it always is the case. Thus, even if many of my comments and suggestions were not accepted by the authors, regarding my initial concerns, I believe that the paper is ready for publication.